# Fatty Acid Binding Protein 5 Mediates Cell Death by Psychosine Exposure through Mitochondrial Macropores Formation in Oligodendrocytes

**DOI:** 10.3390/biomedicines8120635

**Published:** 2020-12-20

**Authors:** An Cheng, Ichiro Kawahata, Kohji Fukunaga

**Affiliations:** Department of Pharmacology, Graduate School of Pharmaceutical Sciences, Tohoku University, Sendai 980-8578, Japan; cheng.an.q6@dc.tohoku.ac.jp (A.C.); kawahata@tohoku.ac.jp (I.K.)

**Keywords:** FABP5, psychosine toxicity, oligodendrocyte, VDAC-1, BAX, mitochondrial pore, apoptosis

## Abstract

Oligodendrocytes, the myelinating cells in the central nervous system (CNS), are critical for producing myelin throughout the CNS. The loss of oligodendrocytes is associated with multiple neurodegenerative disorders mediated by psychosine. However, the involvement of psychosine in the critical biochemical pathogenetic mechanism of the loss of oligodendrocytes and myelin in krabbe disease (KD) remains unclear. Here, we addressed how oligodendrocytes are induced by psychosine treatment in both KG-1C human oligodendroglial cells and mouse oligodendrocyte precursor cells. We found that fatty acid binding protein 5 (FABP5) expressed in oligodendrocytes accelerates mitochondria-induced glial death by inducing mitochondrial macropore formation through voltage-dependent anion channels (VDAC-1) and BAX. These two proteins mediate mitochondrial outer membrane permeabilization, thereby leading to the release of mitochondrial DNA and cytochrome C into the cytosol, and the activation of apoptotic caspases. Furthermore, we confirmed that the inhibition of FABP5 functions by shRNA and FABP5-specific ligands blocking mitochondrial macropore formation, thereby rescuing psychosine-induced oligodendrocyte death. Taken together, we identified FABP5 as a critical factor in mitochondrial injury associated with psychosine-induced apoptosis in oligodendrocytes.

## 1. Introduction

Oligodendrocytes are myelinating glial cells in the central nervous system (CNS) that constitute about 5–10% of the total glial population, and are critical for rapid action potential propagation and metabolic support of neurons [1,2]. The production and remodeling of myelin continues through adulthood, and novel myelin sheath synthesis even plays a vital role in learning [3]. Loss of oligodendrocytes accounting for myelin in the adult CNS is a hallmark of demyelinating diseases such as multiple system atrophy (MSA), multiple sclerosis (MS), and krabbe disease (KD). KD is a rare neurodegenerative disorder caused by a deficiency of the enzyme β-galactosyl-ceramidase (GALC), accumulation of glycosphingolipids (GSL), and loss of myelin-forming cells. The accumulation of psychosine is recognized to be a critical biochemical pathogenetic mechanism of the loss of oligodendrocytes and myelin in KD [4]. As reported previously, psychosine mediates several toxic effects via mitochondria, including inhibition of the electron transport chain, oxidative phosphorylation, and calcium transport [5], thereby triggering cytochrome C release and loss of mitochondrial membrane potential, which in turn results in oligodendrocyte apoptosis [6]. Recent studies have also demonstrated the potential effect of psychosine accumulation in lipid rafts [7] and on disruption of membrane integrity in mouse primary oligodendrocytes [8] through reducing the compactness of the hydrophobic portion, making the membrane more easily compressed [9]. However, the mechanism by which psychosine triggers mitochondrial dysfunction and induces oligodendrocyte death remains unclear.

Fatty acid binding proteins (FABPs) are members of a superfamily and are essential for mitochondrial energy production and gene expression by fatty acids and other lipophilic substances including eicosanoids and retinoids [10,11]. Among them, FABP3, FABP5, and FABP7 are abundantly expressed in the brain. Additionally, they also mediate various neurodegenerative disorders such as Parkinson’s disease (PD) [12,13,14] and MS [15,16]. Importantly, FABP5 inhibition suppresses IL-17 cytokine production and skews T cells toward a Treg phenotype in regulatory T cells (Tregs) [17]. This is consistent with the observation that FABP5 inhibition causes reduced clinical symptoms of experimental autoimmune encephalomyelitis (EAE) in mouse models. [15]. On the other hand, FABP5 maintains mitochondrial integrity, and FABP5 inhibition in Tregs alters mitochondria and suppresses Treg activities [17]. Considering the tight connection between psychosine and FABP5 in mitochondrial pathophysiology, it is worthwhile to confirm how FABP5 works in mitochondria after psychosine exposure in oligodendrocytes.

Mitochondria are vital organelles for brain development and synapse functions, and their importance has been described in many neurodevelopmental and neurodegenerative processes [18]. In fact, impairment of the respiratory chain in mitochondria is a key feature in sporadic PD patients and the proteins encoded by PD-associated genes are linked to disturbances in mitochondrial function in PD patients as well as in chronic MPTP/probenecid mouse models [19]. In lysosomal storage diseases (LSD), including KD, altered mitochondrial mass, morphologic abnormalities, and functional deficits have been reported to be increased, likely secondary to a block in lysosome-dependent degradation [20]. On the other hand, the voltage-dependent anion channel 1 (VDAC-1) is a vital outer mitochondrial membrane protein that transports ATP, ions and other metabolites between the mitochondria and cytosol [21]. However, during apoptosis, the N-terminal domain of VDAC1 interacts with mtDNA, promoting VDAC1 oligomerization and forming mitochondrial pores to release mtDNA fragments in mouse embryonic fibroblasts [22]. Consistent with VDAC-1, BAK/BAX macropores also facilitate mtDNA and cytochrome C efflux and form part of the apoptosome complex, which successively activates caspase-9 and the apoptotic effector caspases, caspase-3 and caspase-7, in mouse embryonic fibroblasts [23].

In the present study, we focused on the critical role of FABP5 in eliciting mitochondria-mediated apoptosis by inducing mitochondrial macropore formation with VDAC-1 and BAX. We found that mitochondrial macropore formation triggers the release of cytochrome C and mtDNA in oligodendrocytes after treatment with psychosine. Importantly, FABP5 inhibition by shRNA and ligands markedly blocked mtDNA/cytochrome C release and rescued the oligodendrocytes from apoptosis. Thus, FABP5 inhibition represents a potential therapeutic approach for treating diseases associated with psychosine toxicity.

## 2. Experimental Section

### 2.1. Cell Culture

KG-1C human oligodendroglial cells were obtained from RIKEN BRC Cell Bank (Tsukuba, Ibaraki, Japan). KG-1C human oligodendroglial cells were cultured in Dulbecco’s minimal essential medium (DMEM) containing 10% fetal bovine serum (FBS) and penicillin/streptomycin (100 units/100 µg/mL) at 37 °C under 5% carbon dioxide (CO_2_). Arachidonic acid (AA) (prepared as described previously [13]), lipopolysaccharide (LPS) (dissolved in DMSO), and psychosine (dissolved in DMSO) were used at concentrations of 100 μM, 10 μg/mL, and 5 μM, respectively. 

### 2.2. Animals and Oligodendrocyte Precursor Cell Culture

Pregnant C57BL/6J mice were obtained from Clea Japan, Inc. (Tokyo, Japan), and housed in polypropylene cages (temperature: 23 ± 2 °C; humidity: 55 ± 5%; lights on between 9 a.m. and 9 p.m.). For oligodendrocyte precursor cell culture, P1–2 mouse pups were decapitated and cortex from the brains was dissected and digested in a digestion solution (13.6 mL PBS, 0.8 mL DNase I stock solution (0.2 mg/mL), and 0.6 mL of a trypsin stock solution (0.25%)). Cells were then diced using a sterilized razor blade into ~1 mm^3^ chunks, centrifuged at 100× *g* for 5 min, and resuspended in DMEM20S medium (DMEM, 4 mM l-glutamine, 1 mM sodium pyruvate, 20% FBS, 50 U mL^−1^ penicillin, and 50 µg/mL streptomycin). The tissue suspension was strained using a 70 µm nylon cell strainer and seeded in poly l-lysine-coated tissue culture flasks and cultured in DMEM20S medium. After 10 days, the flasks were shaken for 1 h at 200 rpm at 37 °C to remove microglial cells, and for an additional 20 h to detach OPCs, and then seeded on poly d-lysine-coated plates and cultured in OPC medium (DMEM, 4 mM l-glutamine, 1 mM sodium pyruvate, 0.1%BSA, 50 µg/mL Apo-transferrin, 5 µg/mL insulin, 30 nM sodium selenite, 10 nM d-biotin, 10 nM hydrocortisone, 10 ng/mL PDGF-AA, and 10 ng/mL bFGF). Ethical approval was obtained from the Institutional Animal Care and Use Committee of the Tohoku University Environmental and Safety Committee (2019PhLM0-021 and 2019PhA-024).

### 2.3. Protein Extraction

Cultured cells were frozen in liquid nitrogen and stored at −80 °C. Frozen cells were homogenized with 50 µL of Triton X-100 buffer (0.5% Triton-X100, pH 7.4, 4 mM ethylene glycol [EGTA]), 50 mM Tris-HCl, 10 mM ethylenediaminetetraacetic acid (EDTA), 1 mM sodium orthovanadate (Na_3_VO_4_), 50 mM sodium fluoride (NaF), 40 mM Na_4_P_2_O_7_·10H_2_O, 0.15 M sodium chloride (NaCl), 50 µg/mL leupeptin, 25 µg/mL pepstatin A, 50 µg/mL trypsin inhibitor, 100 nM calyculin A, and 1 mM dithiothreitol) for each 35 mm dish. Concentrations of the supernatant protein were normalized using a Bradford assay.

### 2.4. Mitochondria Isolation

Mitochondria were isolated from KG-1C cells as previously described [24]. Briefly, cells were suspended in mitochondrial isolation buffer containing 250 mM sucrose, 1 mM dithiothreitol, 10 mM KCl, 1 mM EDTA, 1.5 mM MgCl_2_, protease inhibitors and 20 mM Tris-HCl, pH 7.4, then homogenized using a glass homogenizer with approximately 50 strokes per pestle, and centrifuged twice at 800× *g* for 10 min. The supernatants were collected and centrifuged at 15,000× *g* for 10 min at 4 °C. Supernatants were collected as cytosolic fractions (without mitochondria). The mitochondrial pellets were immediately washed three times with mitochondrial isolation buffer, homogenized with Triton X-100 buffer as described above, and the supernatant was collected as mitochondrial fractions. Protein concentrations of the isolated mitochondria were normalized using a Bradford assay. The quantity of mitochondria was estimated by VDAC-1 using Western blotting.

### 2.5. Immunoblotting Analysis 

The extracts from cells or isolated mitochondria were separated by SDS-polyacrylamide gel electrophoresis (SDS-PAGE) with a ready-made gel (Cosmo Bio Co., Ltd., Tokyo, Japan) and transferred to polyvinylidene difluoride (PVDF) membranes. The membranes were incubated with primary antibodies against FABP7, FABP5, MFN-2, VDAC-1, β-tubulin, BAX, dsDNA, cytochrome C, IL-1β, and Cleaved Caspase-3, followed by treatment with a horseradish peroxidase (HRP)-conjugated secondary antibody (anti-mouse IgG (H&L), anti-rabbit IgG (H&L) and anti-goat IgG (H&L)) (Table 1), and protein was detected using an ECL detection system (Amersham Biosciences, Buckinghamshire, UK) by Image Quant LAS 4000mini system. Intensity quantification was conducted using Image Gauge software version 3.41 (Fuji Film, Tokyo, Japan).

### 2.6. Immunoprecipitation Analysis

For the immunoprecipitation analysis, 50 µL of protein A-Sepharose CL-4B (50%, *v*/*v*) was suspended in phosphate-buffered saline (PBS) in a total volume of 500 µL and stored at 4 °C. Cell extracts containing 200 µg of protein were incubated for 2 h at 4 °C with 10 µg of anti-FABBP5 antibody, 5 µg of anti-VDAC-1 antibody or anti-BAX antibody. The mixture was incubated at 4 °C for at least 4 h. Samples were then separated using SDS-PAGE with a ready-made gel (Cosmo Bio Co., Ltd.).

### 2.7. Sucrose Gradient Centrifugation 

To measure the molecular weight shift of VDAC-1 following psychosine exposure, we used sucrose gradient centrifugation. The sucrose densities were 55%, 40%, 30%, and 20% in PBS. Cell lysates were centrifuged at 100,000× *g* for 20 h at 4 °C. Fractions were analyzed by Western blotting with an anti-VDAC-1 antibody (1:500; 4866; CST, Tokyo, Japan), an anti-BAX antibody (1:500; SC-493; Santa Cruz, Dallas, TX, USA), and an anti-FABBP5 antibody (R&D Systems, AF3077, Minneapolis, MN, USA).

### 2.8. Dot Blot Assay

To determine dsDNA levels in the mitochondrial fraction, a dot blot assay was performed as previously described [25]. Briefly, a PVDF membrane was placed on the top of the soaked sheets and equal amounts of protein in a similar volume were placed in dots in specific zones. After drying, the membrane was blocked with 5% nonfat milk in TTBS buffer (0.1% Tween 20 in Tris-buffered saline) for 1 h at room temperature. Membranes were then incubated with primary antibodies against ds-DNA and VDAC-1, followed by treatment with a horseradish peroxidase (HRP)-conjugated secondary antibody (anti-mouse IgG (H&L) and anti-rabbit IgG (H&L)). Images were detected using an ECL detection system (Amersham Biosciences, Buckinghamshire, UK) by Image Quant LAS 4000mini system. Intensity quantification was conducted using Image Gauge software version 3.41 (Fuji Film, Tokyo, Japan).

### 2.9. Immunofluorescent Staining and Confocal Microscopy

Immunofluorescent staining was conducted as previously described [26]. In the present study, cells were incubated with primary antibodies against FABP7, FABP5, TOM20, Olig2, and dsDNA. Fluorescein, Alexa 405-labeled anti-mouse IgG; Alexa 488-labeled anti-goat IgG; Alexa 594-labeled anti-mouse IgG; and Alexa 594-labeled anti-rabbit IgG were used for detection. Mito-Tracker (CST) and Alexa Fluor™ 488 Phalloidin (Thermo Fisher Scientific, Waltham, MA, USA) were used for mitochondria and filamentous actin staining, respectively. Immunofluorescent images were analyzed using a confocal laser scanning microscope (DMi8; Leica, Wetzlar, Germany).

### 2.10. Cell Death Assay

Cell viability was measured using a cell counting kit (CCK; Dojindo, Japan), according to the manufacturer’s instructions. Absorbance by viable cells was measured at a test wavelength of 400 nm and a reference wavelength of 450 nm using a FlexStation 3 Multi-Mode Microplate Reader (Molecular Devices, San Jose, CA, USA).

### 2.11. FABP5 shRNA Plasmid

Human-FABP5 shRNA Bacterial stock was obtained from Sigma-Aldrich, and the sequence was as follows: shRNA1 (CCGGGTGGAGTGTGTCATGAACAATCTCGAGATTGT TCATGACACACTCCACTTTTTTG); shRNA2 (CCGGGCAACTTTACAGATGGTGC ATCTCGAGATGCACCATCTGTAAAGTTGCTTTTTG), shRNA3(CCGGTGAGCAAA TCTCCATACTGTTCTCGAGAACAGTATGGAGATTTGCTCATTTTTTG); shRNA4 (CCGGCTGGGAGAGAAGTTTGAAGAACTCGAGTTCTTCAAACTTCTCTCCCAGTTTTTG). The plasmid was purified using the GenElute^TM^ HP Plasmid Maxiprep Kit (Sigma, St. Louis, MO, USA).

### 2.12. shRNA Delivery

KG-1C cells were transduced with an empty vector or FABP5 shRNA plasmid (2 μg per 35 mm dish) using Lipofectamine LTX and Plus Reagent (Invitrogen, Carlsbad, CA, USA) and Opti-MEM (Thermo Fisher Scientific, Waltham, MA, USA), according to the manufacturer’s protocol.

### 2.13. Analysis of Mitochondrial Membrane Potential (JC-1 Assay)

Treated KG-1C cells were washed with PBS and stained with JC-1 Mito MP Detection Kit (Dojindo), according to the manufacturer’s instructions. Relative degrees of mitochondrial polarization were quantified by measuring the red-shifted JC-1 aggregates at 535 nm (Ex), 585–605 nm (Em), and green-shifted JC-1 aggregates at 485 nm (Ex) and −525–545 nm (Em) using FlexStation 3 Multi-Mode Microplate Reader (Molecular Devices, San Jose, CA, USA).

### 2.14. FABP5 Ligands

The FABP5 ligands were synthesized by SHIRATORI PHAMACEUTICAL CO., LTD as described previously [13].

### 2.15. Statistical Analysis

Data are expressed as the mean ± standard error of the mean (SEM). Statistical comparisons were evaluated using one-way analysis of variance (ANOVA), followed by Tukey’s multiple comparisons test, as required. A *p* value of < 0.05 was considered statistically significant.

## 3. Results

### 3.1. Psychosine Triggered Mitochondria Loss and Abnormal Accumulations of FABP5 in Mitochondria

Previous studies have indicated that psychosine mediates toxicity effects, especially in oligodendrocytes [4], and causes direct cellular cytotoxicity by inducing mitochondrial dysfunction [6]. To further understand the potential relationship between psychosine toxicity and mitochondrial dysfunction, we treated KG-1C cells, which express endogenous FABP5 and FABP7, with psychosine at low concentrations (5 μM). As a result, we found significant degeneration of KG-1C cells in terms of dendrite length (*p* < 0.01, *n* > 40) (Figure 1A,C) and a decreased number of mitochondria in dendrites (*p* < 0.01, *n* > 26), but no change in the number of mitochondria in cell bodies was observed after the treatment (Figure 1B,D). We also treated KC-1C cells with AA and LPS which are widely used in triggering oxidative stress and neuroinflammation. However, both AA (100 μM) and LPS (10 μg/mL) failed to induce degeneration of KG-1C cells in terms of dendrite length and loss of mitochondrial numbers (Figure 1A–D). Furthermore, to understand the morphological changes in mitochondria due to psychosine toxicity, we stained mitochondria using TOM20 and Mito-tracker, and observed a pronounced mitochondrial fission that was induced by psychosine in a time-dependent manner (Figure 1E). Importantly, the mitochondrial fission process induced by psychosine was associated with FABP5 migration into mitochondria characterized by TOM20 and Mito-tracker staining (Figure 1E–G). Likewise, we observed a significant increase in FABP5 in mitochondrial fractions after treatment with psychosine for 24 h (*p* < 0.01, *n* = 4; Figure 1H,J). However, we observed no such changes in FABP7 after treatment with psychosine (Figure 1H,J; Appendix A). This further verifies the changes in FABP5 located in mitochondria under psychosine stress in psychosine-treated OPCs. Consistent with the results in KG-1C cells, FABP5 was localized in the mitochondria of OPCs after psychosine treatment (Figure 2D,E). Taken together, FABP5 migrates into the mitochondrial compartment during mitochondrial fission following psychosine cytotoxicity in oligodendrocytes. 

### 3.2. FABP5 Engaged in Mitochondrial Pores Formation under Psychosine Stress

Mitochondrial dysfunction is known to be mediated by macropore formation due to increased expression of VDAC-1 [22] and BAX [23] during mitochondria-induced apoptosis. In the present study, we also confirmed an elevation in outer mitochondrial membrane proteins including VDAC-1 and BAX in psychosine-treated KG-1C cells. VDAC-1 oligomers, especially those with high molecular weight multimers, increased in mitochondrial fractions after a 24 h psychosine treatment (*p* < 0.01, *n* = 4) (Figure 3A,B), indicating that psychosine triggers mitochondrial pore formation. On the other hand, the levels of BAX proteins with high molecular weight multimers did not change. Interestingly, when we incubated the membrane with the FABP5 antibody after stripping the VDAC-1, FABP5 levels were also elevated in high molecular weight fractions in a time-dependent manner (Figure 3C). In this context, FABP5 likely binds to VDAC-1 and forms mitochondrial pores with VDAC-1. To confirm our hypothesis, we performed an immunoprecipitation analyses and a sucrose gradient. Both VDAC-1 and BAX were found in the FABP5 antibody-immunoprecipitated fraction and FABP5-positive fractions in the sucrose gradient (Figure 3D). Inversely, a FABP5-immunoreactive band was detected in both the VDAC-1 and BAX precipitated fractions (Figure 3E). As expected, the psychosine treatment had no effect on VDAC-1 and FABP5 interaction in immunoprecipitation analyses. Consistent with these results, both FABP5 and VDAC-1 fractions were shifted to more heavy sucrose fractions including fractions 14 and 15 after the psychosine treatment. However, BAX slightly shifted to fractions 9 and 10 (Figure 3F,G). These results indicate that FABP5 participates in mitochondrial pore formation with VDAC-1 oligomerization. It has been proposed that VDAC-1 oligomer-dependent mitochondrial pores mediate fragmented mtDNA release [22]. Based on previous reports, we confirmed the release of mtDNA after psychosine treatment. As a result, mtDNA is normally located in mitochondria surrounded by the mitochondrial outer membrane protein TOM20 in control cells (Figure 3H). In psychosine-treated cells, mtDNA was released into the cytosol from the mitochondria, where FABP5 migrated to the mitochondria with TOM20 (Figure 3H). Consistent with the confocal microscopy results, in dot blot analyses, ds-DNA levels were significantly decreased in the mitochondrial fraction following psychosine treatment (*p* < 0.01; *n* = 4; Figure 3I,J); this result may be related to mitochondrial pore formation induced by VDAC-1 and FABP5 following psychosine treatment. Taken together, FABP5 participated in BAX and VDAC-1 oligomer-dependent mitochondrial pore formation and induced mtDNA release.

### 3.3. FABP5 Is Essential in Psychosine-Induced Mitochondrial Pores Formation

We further confirmed the function of FABP5 in psychosine-induced oligodendrocyte degeneration, and found that FABP5 expression levels were decreased by FABP5 shRNA (Sigma-Aldrich). The shRNA 3 treatment was the most efficient (Figure 4A) for FABP5 deletion. The shRNA 3 treatment rescued cell death after psychosine treatment (*p* < 0.01, *n* = 8; Figure 4B) and recovered mitochondrial membrane potential (*p* < 0.01, *n* = 6; Figure 4C). These results suggest that the psychosine toxicity towards mitochondria is mediated by FABP5. We also confirmed the presence of VDAC-1 oligomer formation with a 60–210 kDa decrease in the mitochondrial fraction in shRNA 3 treated cells (*p* < 0.01, *n* = 6; Figure 4D,E). Since VDAC-1 oligomer formation under oxidative stress accelerates the release of cytochrome C, causing apoptotic caspase activation [22,27], we further confirmed the increased levels of cytosolic cytochrome C, IL-1β, and cleaved caspase-3 in the cytosolic fraction by Western blotting. In addition to cytochrome C, the levels of IL-1β and cleaved caspase-3 were significantly increased in psychosine-treated cells (*p* < 0.01, *n* = 4). The shRNA 3 treatment significantly attenuated the release of cyto chrome C (*p* < 0.05, *n* = 4), IL-1β (*p* < 0.01, *n* = 4), and levels of cleaved caspase-3 (*p* < 0.01) in the cytosolic fraction (Figure 4D,F–H). Taken together, deletion of FABP5 using shRNA attenuated the psychosine toxicity towards mitochondria and rescued oligodendrocytes from death.

### 3.4. Pharmacological Inhibition of FABP5 in Psychosine Toxicity

Since we found that psychosine toxicity towards mitochondria required FABP5, it is necessary to define the effect of pharmacological inhibition of FABP5 by FABP5 inhibitors on psychosine toxicity. We recently introduced several FABP inhibitors, four of which—ligand 1 (Kd = 324 ± 45 nM), ligand 6 (Kd = 874 ± 66 nM), ligand 7 (Kd = 199 ± 23 nM), and ligand 8 (Kd = 324 ± 39 nM)—elicit a high binding affinity for FABP5 (Figure 5B) [28]. Among them, ligand 6 (*p* < 0.01, *n* = 8) and ligand 8 (*p* < 0.01, *n* = 8) elicited the greatest amelioration of cell death (Figure 5C). On the other hand, in the JC-1 assay, ligand 7 was the most effective in rescuing the mitochondrial membrane potential (*p* < 0.01, *n* = 8) (Figure 5D). To confirm rescuing the mitochondrial dysfunction, we treated KG-1C cells with ligand 7 and confirmed that ligand 7 decreased VDAC-1 oligomer formation (*p* < 0.01, *n* = 4), release of cytosolic cytochrome C (*p* < 0.05, *n* = 4), and IL-1β (*p* < 0.01, *n* = 4), and cleaved caspase-3 levels (*p* < 0.01, *n* = 4) (Figure 5E–I). Taken together, the inhibition of FABP5 by FABP5 inhibitors significantly inhibited mitochondrial pore formation and rescued oligodendrocytes from apoptosis.

## 4. Discussion

It has been proposed that psychosine is a cationic lysosphingolipid that accumulates in the brain of patients with KD [6] and causes oligodendrocyte cell death and demyelination [29]. Compared with hippocampal and cortical neurons, primary oligodendrocytes are known to be more sensitive to psychosine [30]. Since psychosine induces oligodendrocyte loss in patients with KD, it is important to determine the mechanism(s) underlying psychosine toxicity. Consistent with a previous study, we found that psychosine mediates several toxic effects in mitochondria, such as decreased mitochondrial numbers in KG-1C cells and OPCs. On the other hand, no significant changes in mitochondrial number were observed after treatment with AA and LPS. This may indicate a tight association between psychosine toxicity and mitochondrial dysfunction in oligodendrocytes. Importantly, during mitochondrial fission induced by psychosine, FABP5 but not FABP7 migrates into the mitochondrial fraction. In addition, FABP5 is critical for trafficking lipids to mitochondria in mammals [31], and FABP5 acts as a gatekeeper of mitochondrial integrity by regulating lipid metabolism in Tregs [17]. Taken together, FABP5 is also essential for mitochondrial function in oligodendrocytes. 

Since mitochondrial pore formation has been reported to mediate cell apoptosis [32], we focused on two main mitochondrial membrane proteins, VDAC-1 and BAX, which mediate mitochondrial pore formation in mouse embryonic fibroblasts [22,23]. Importantly, FABP5 participates in VDAC-1 oligomer-dependent mitochondrial pore formation. Since mitochondrial inner membrane permeabilization occurs during cell death with BAX/BAK and VDAC-1 oligomer-dependent mitochondrial outer membrane permeabilization [22,32], we confirmed the decrease in ds-DNA levels in mitochondrial fractions after the treatment. The mtDNA, but not nuclear DNA, is particularly susceptible to attack by H_2_O_2_ due to lack of protective histones [33]. Recent findings also indicated that mtDNA release triggers the activation of the cGAS-CGMP-STNG pathway in adipocytes [34], causing dying cells to secrete type 1 interferon. Moreover, mitochondrial dysfunction specifically triggers IL-1β secretion, and cytosolic release of mtDNA is associated with NLRP3 inflammasome-dependent apoptosis in macrophages [35]. This is consistent with our finding that IL-1β levels increased after treatment with psychosine. Furthermore, we also found elevated cytosolic cytochrome C levels in psychosine-treated cells when we treated them with a higher concentration of psychosine (10 μM). Normally, cytochrome C resides in the inner membrane of the mitochondria and serves as a component of the electron transport chain [36,37]. However, when cytochrome C is released into the cytosol, it activates cysteine proteases including caspase 9, caspase 7, and caspase 3 and eventually causes cell apoptosis [38,39,40]. Consistent with previous reports, we observed an approximately twofold increase in cleaved caspase-3 in the cytosolic fraction. 

To address the question of whether the toxicity of psychosine in mitochondria is regulated by FABP5, we inhibited FABP5 by shRNA- and FABP5-specific inhibitors. Despite a slightly worse viability in shRNA-treated cells, FABP5 partial deletion by shRNA and treatment with FABP5 inhibitors significantly rescued oligodendrocytes from psychosine-induced cell death. Therefore, FABP5 is a critical factor in the mitochondrial injury associated with psychosine toxicity in oligodendrocytes. We next tested the FABP5 inhibitors in a mouse model of KD to determine its function in vivo, as well as the glial pathology associated with KD. 

The present study could not reveal specifically which membrane pore (BAX/BAK or VDAC-1) is mediated by FABP5 that leads to the dysregulation of mitochondrial pore formation under psychosine stress. We confirmed that FABP5 forms complexes with VDAC-1 and BAX by immunoprecipitation and formed high molecular weight complexes with VDAC-1 in the sucrose gradient. In the sucrose density gradient, FABP5 failed to produce an oligomer with BAX after the psychosine treatment. Previous studies have proposed that anti-VDAC antibodies inhibit cytochrome C release and the interaction with Bax during cell death [41,42,43]. Overexpression of VDAC-1 elicits apoptotic cell death in U-937 human monocytic cells [44,45]. Moreover, VDAC can oligomerize under oxidative stress and form large mitochondrial outer membrane pores [27] to release mtDNA [21]. On the other hand, BAX/BAK oligomers can form extremely large mitochondrial pores and have been shown to mediate mtDNA and cytochrome C release under conditions that activate BAX/BAK activators [23,46,47]. Although whether psychosine activates BAX/BAK activators remains unclear in the present study, we observed that both mtDNA and cytochrome C release to the cytosol in psychosine-treated cells and this was partly mediated by FABP5. Overall, FABP5 facilitates mitochondrial macropore formation and induces cytochrome C/mtDNA-related apoptosis under psychosine toxicity in oligodendrocytes.

In conclusion, we provide the first evidence that FABP5 mediates psychosine toxicity by inducing mitochondrial pore formation with BAX and VDAC-1 oligomers in oligodendrocytes. Mitochondrial pore formation likely triggers the release of mtDNA and cytochrome C, thereby accelerating psychosine-induced cell death in oligodendrocytes. Furthermore, FABP5 inhibition by shRNA and specific ligands significantly blocks the toxicity of psychosine and rescues oligodendrocytes (Figure 6). Therefore, inhibition of FABP5 represents a potential therapeutic approach for diseases associated with psychosine toxicity, such as KD.

## Figures and Tables

**Figure 1 biomedicines-08-00635-f001:**
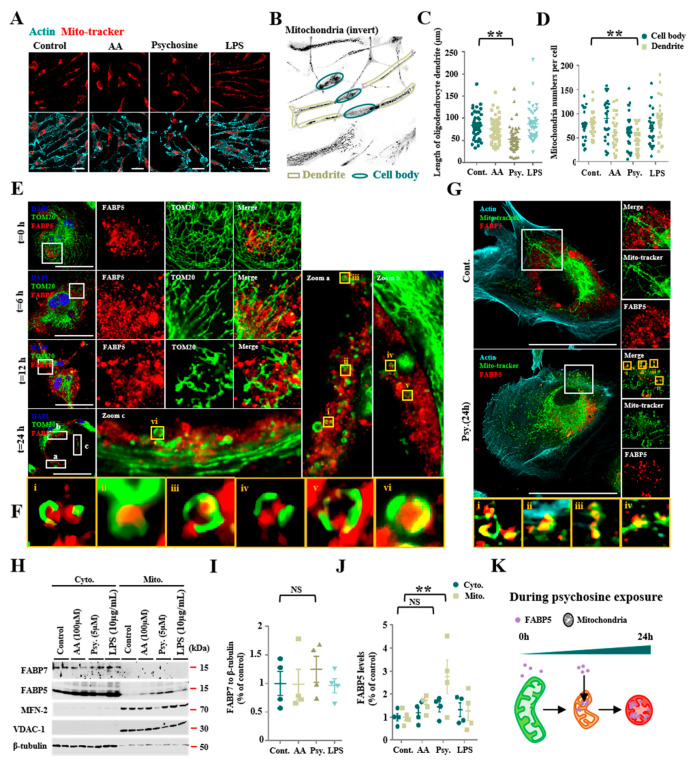
Abnormal accumulation of FABP5 in mitochondria under psychosine exposure. (**A**) Confocal images of actin (cyan) and Mito-tracker (red), KG-1C cells were treated with AA (100 μM), psychosine (5 μM), or LPS (10 μg/mL), respectively. (**B**) Microscopy image of KG-1C cells, stained with Mito-tracker. Inverted image was processed by ImageJ software. (**C**,**D**) Quantification of dendrite length (*n* > 40) (C) and mitochondria numbers in the cell body and dendrites (*n* > 26) (**D**). KG-1C cells were treated with AA (100 μM), psychosine (5 μM), or LPS (10 μg/mL), respectively. (**E**) Confocal images of TOM20 (green), FABP5 (red), and DAPI (blue). KG-1C cells were treated with psychosine (5 μM) for 0, 6, 12, or 24 h, respectively. Mitochondrial fission occurs during psychosine exposure and FABP5 shifts to mitochondria during mitochondrial fission. (**F**) Amplified images of mitochondria in (**E**), noise was processed by waifu2x-caffe software. (**G**) Confocal images of Mito-tracker (green), FABP5 (red), and actin (cyan). KG-1C cells treated with psychosine (5 μM) for 24 h, show co-localization of FABP5 and mitochondria after psychosine treatment. (**H**) Western blot analysis showing that FABP5, but not FABP7, is elevated in the mitochondrial fraction in psychosine-treated KG-1C cells (*n* = 4). (**I**,**J**) Quantification of FABP7 in the cytosolic fraction and FABP5 in both the cytosolic fraction and mitochondrial fraction. (**K**) Schematic diagram of FABP5 shifting to mitochondria during psychosine exposure. The data are shown as mean ± standard error of the mean and were obtained using a one-way analysis of variance. ** *p* < 0.01. AA, arachidonic acid; psy, psychosine; LPS, lipopolysaccharide; Cyto., cytosolic fraction; Mito., mitochondrial fraction; NS, not statistically significant. Scale bar = 50 μm.

**Figure 2 biomedicines-08-00635-f002:**
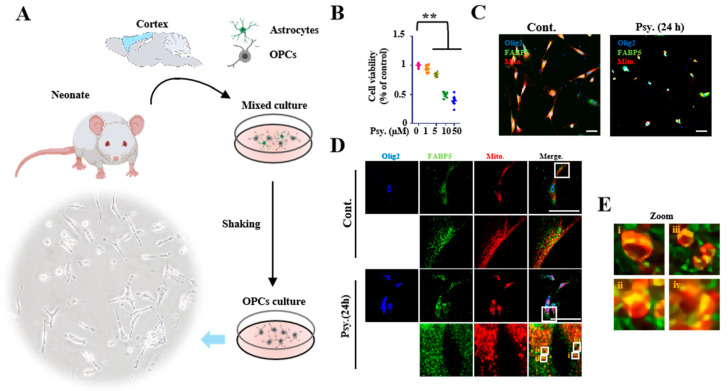
FABP5 localized in mitochondria of OPCs. (**A**) Schematic draft of the primary culture protocol for OPCs. OPCs were isolated from the cortex of neonates and purified by shaking and selective culture. (**B**) Cell viability analysis of OPCs, based on CCK assay. OPCs were treated with psychosine at various concentrations. (**C**) Confocal microscopy of immunofluorescence staining of olig2 (blue), FABP5 (green), Mito-tracker (red). OPCs were treated with psychosine (10 μM) for 24 h. (**D**) Confocal microscopy of OPCs showing FABP5 co-localized with mito-tracker after psychosine treatment. (**E**) Amplified images of mitochondria in (**D**), noise was processed by waifu2x-caffe software (https://github.com/lltcggie/waifu2x-caffe/releases). The data are shown as mean ± standard error of the mean and were obtained using a one-way analysis of variance. ** *p* < 0.01. Cont., control; psy, psychosine; OPCs, oligodendrocyte precursor cells. Scale bar = 100 μm.

**Figure 3 biomedicines-08-00635-f003:**
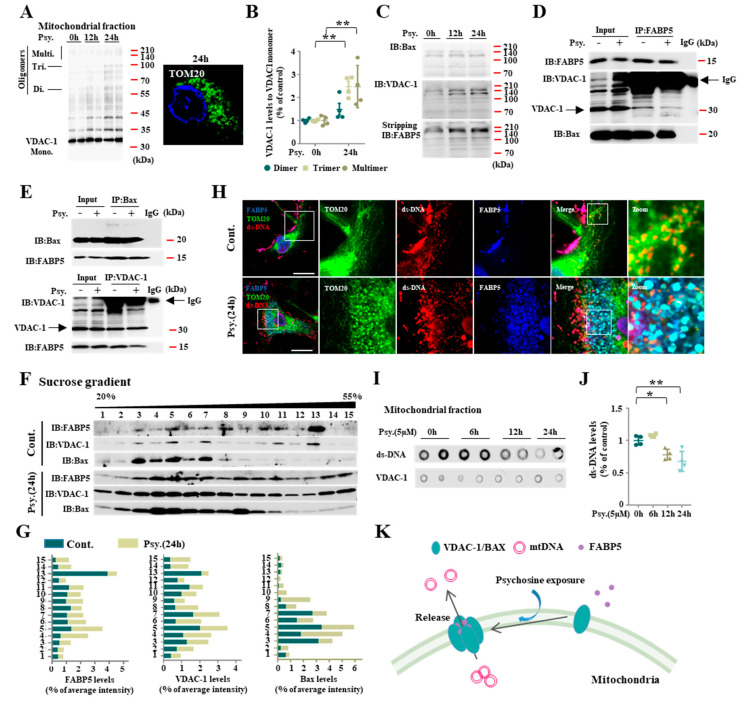
FABP5 facilitated mitochondria macropore formation following psychosine exposure. (**A**) Western blot analysis of VDAC-1 in mitochondrial fraction (left), showing that VDAC-1 dimers (60 kDa), trimers (about 100 kDa), multimers (above 140 kDa) appeared in psychosine-treated KG-1C cells. Confocal microscopy of TOM20 (green) and DAPI (blue) (right), showing mitochondrial loss after psychosine treatment. (**B**) Quantification of VDAC-1 levels, suggests an increase in VDAC-1 oligomers in psychosine-treated KG-1C cells (*n* = 4). (**C**) Western blot analysis of BAX, VDAC-1 and FABP 5 in high molecular weight, showing potential effects of FABP5 in VDAC-1 oligomerization. (**D**,**E**) Co-immunoprecipitation of FABP5, BAX and VDAC-1, suggests that FABP5 makes complexes with VDAC-1 and BAX. (**F**) Sucrose gradient assay of FABP5, VDAC-1 and BAX. Compared with control, in psychosine-treated cells, VDAC-1 and FABP5 shifted to more heavy sucrose fractions (14 and 15), however BAX only shifted a little to fractions 9 and 10. (**G**) Bar graph of (**F**). (**H**) Confocal microscopy of immunofluorescence staining of ds-DNA (red), TOM20 (green), FABP5 (blue), showing FABP5 shifted to mitochondria and mtDNA released to cytosol from mitochondria. (**I**) Dot blot of dsDNA in the mitochondrial fraction, showing decreased dsDNA in mitochondria after psychosine treatment. (**J**) Quantification of dsDNA in the mitochondrial fraction (*n* = 4). (**K**) Schematic diagram of FABP5 participating in mitochondria pore formation under psychosine exposure. The data are shown as mean ± standard error of the mean and were obtained using a one-way analysis of variance. * *p* < 0.05 and ** *p* < 0.01. Psy, psychosine; IP, immunoprecipitation; IB, immunoblotting. Scale bar = 50 μm.

**Figure 4 biomedicines-08-00635-f004:**
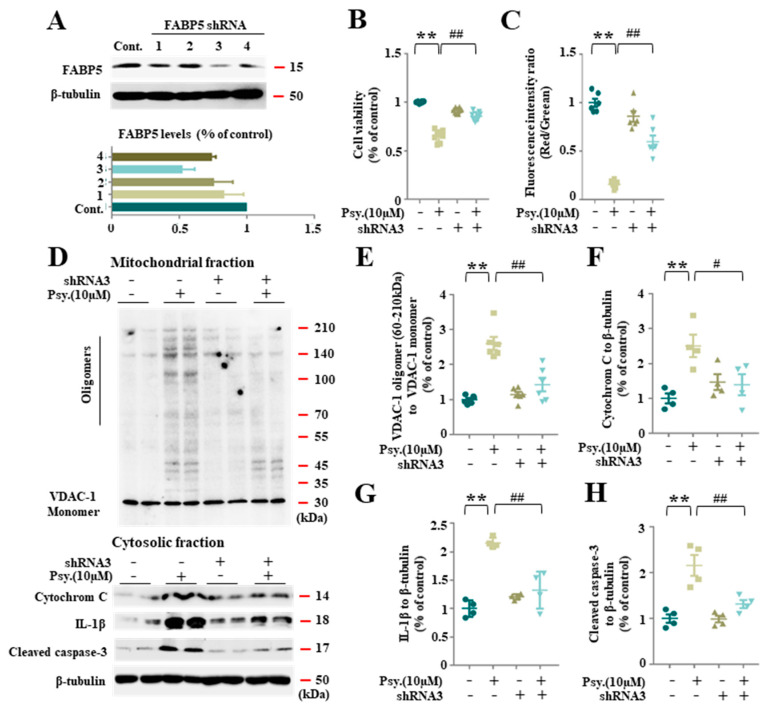
FABP5 inhibition by shRNA reduced mitochondria pore formation. (**A**) Western blot analysis of FABP5 in shRNA transduction KG-1C cells (top), bar graph showing shRNA 3 was the most efficient one (bottom). (**B**) Cell viability analysis of KG-1C, based on CCK assay. KG-1C cells were treated with psychosine (10 μM) or shRNA. We observed a rescue in cell viability in shRNA-treated cells (*n* = 8). (**C**) Analysis of mitochondrial membrane potential by JC-1 assay. KG-1C cells were treated with psychosine (10 μM) or shRNA. We observed a recovered mitochondrial membrane potential in shRNA-treated cells (*n* = 6). (**D**) Western blot analysis of VDAC-1 oligomers in mitochondrial fraction (top), and cytochrome C, IL-1β and cleaved caspase-3 in cytosolic fraction (bottom). (**E**–**H**) Quantification of (**D**), showed decreased VDAC-1 oligomer levels in the mitochondrial fraction in shRNA-treated cells compared to psychosine-treated cells (*n* = 6), and decreased levels of cytochrome C, IL-1β and cleaved caspase-3 in the cytosolic fraction in shRNA-treated cells compared to psychosine-treated cells (*n* = 4). The data are shown as mean ± standard error of the mean and were obtained using a one-way analysis of variance. ** *p* < 0.01; ^#^
*p* < 0.05 and ^##^
*p* < 0.01. Cont., control; psy, psychosine.

**Figure 5 biomedicines-08-00635-f005:**
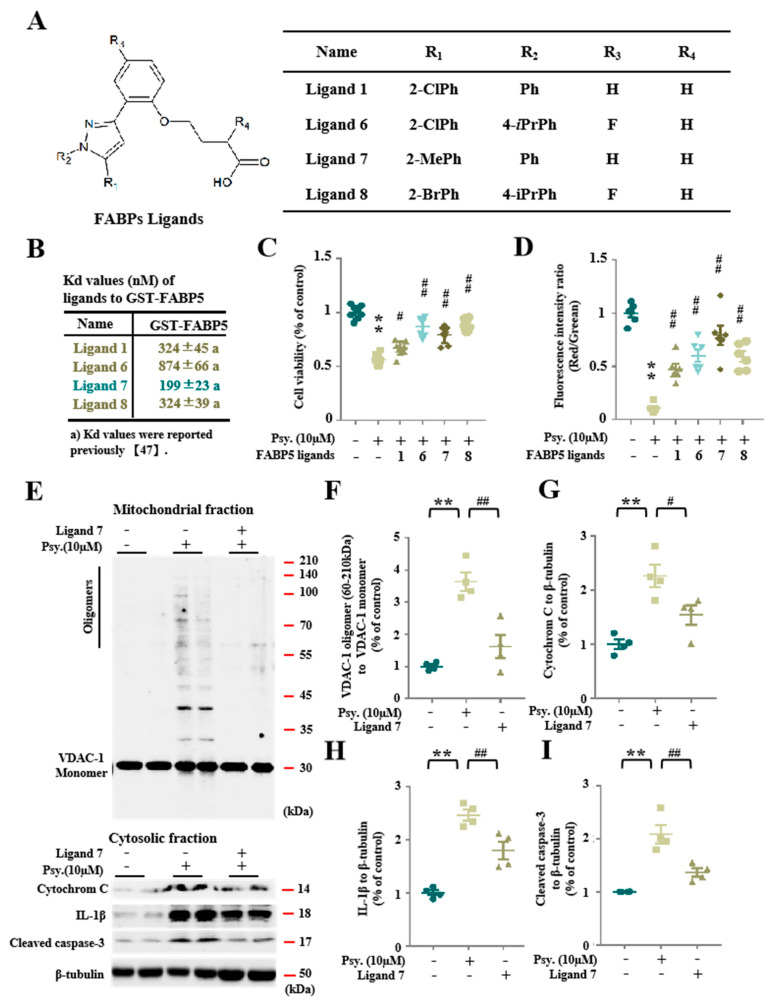
Pharmacological inhibition of FABP5 attenuates psychosine toxicity towards mitochondria. (**A**) Structures of FABP ligands. (**B**) Kd values (nM) of ligands to GST-FABP5, Kd values were reported previously [28]. (**C**) Cell viability analysis of KG-1C, based on the CCK assay. KG-1C cells were treated with psychosine (10 μM) or FABP ligands. Treatment with FABP ligands rescued cell viability, especially in ligand 6- and ligand 8-treated cells (*n* = 8). (**D**) Analysis of mitochondrial membrane potential by JC-1 assay. KG-1C cells were treated with psychosine (10 μM) or FABPs ligands (1 μM). Treatment with FABPs ligands recovered the mitochondrial membrane potential in treated cells, especially in ligand 7-treated cells (*n* = 6). (**E**) Western blot analysis of VDAC-1 oligomers in mitochondrial fraction (top), and cytochrome C, IL-1β and cleaved caspase-3 in cytosolic fraction (bottom). (**F**–**I**) Quantification of (**E**), showing decreased VDAC-1 oligomer levels in the mitochondrial fraction in ligand 7 (1 μM)-treated cells compared to psychosine-treated cells (*n* = 4), and decreased levels of cytochrome C, IL-1β and cleaved caspase-3 in cytosolic fraction in ligand 7 (1 μM)-treated cells compared to psychosine-treated cells (*n* = 4). The data are shown as mean ± standard error of the mean and were obtained using a one-way analysis of variance. ** *p* < 0.01; ^#^
*p* < 0.05 and ^##^
*p* < 0.01. Cont., control; Psy., psychosine.

**Figure 6 biomedicines-08-00635-f006:**
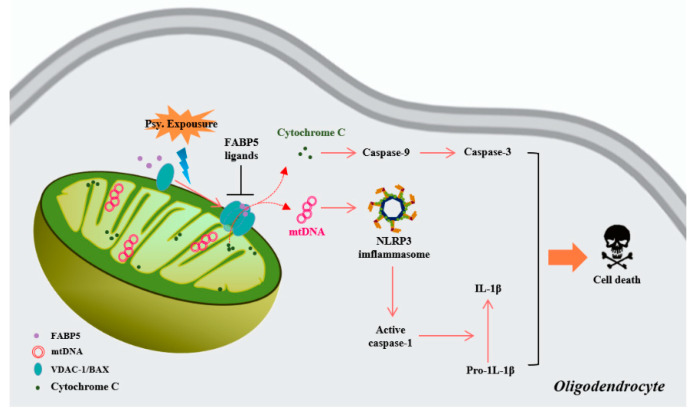
Schematic representation of the pathways through which FABP5 facilitates mitochondrial macropore formation and induces oligodendrocyte apoptosis. During psychosine exposure, FABP5 abnormally accumulates in mitochondria and facilitates mitochondrial macropore formation via VDAC-1 oligomers and BAX. Furthermore, cytosolic cytochrome C and mtDNA released from mitochondrial pores trigger NLRP3 inflammasome-dependent and caspase-dependent apoptosis. In addition, pharmacological inhibition of FABP5 by FABP5 ligands blocks mitochondrial macropore formation, thereby rescuing oligodendrocytes.

**Table 1 biomedicines-08-00635-t001:** Key resources of antibodies.

Designation	Source	Identifiers	Dilution Ratio
FABP7	R&D Systems	AF3166	1:200
FABP5	R&D Systems	AF3077	1:200
MFN-2	abcam	ab56889	1:1000
VDAC-1	CST	4866	1:500
β-tubulin	Sigma-Aldrich	T0198	1:4000
BAX	Santa Cruz	SC-493	1:500
dsDNA	abcam	ab27156	1:1000
Cytochrome C	CST	4272	1:1000
IL-1β	abcam	ab9722	1:1000
Cleaved Caspase-3	CST	9661	1:200
Olig2	Sigma-Aldrich	MABN50	1:500
Anti-mouse IgG (H&L)	SouthernBiotech	1031-05	1:5000
Anti-rabbitIgG (H&L)	SouthernBiotech	4050-05	1:5000
Anti-goatIgG (H&L)	Rockland Immunochemicals	605-4302	1:5000
Alexa 405-labeled ant-mouse IgG	Thermo Fisher Scientific	A-31553	1:500
Alexa 488-labeled ant-goat IgG	Thermo Fisher Scientific	A-11055	1:500
Alexa 594-labeled ant-mouse IgG	Thermo Fisher Scientific	A-21203	1:500
Alexa 594-labeled ant-rabbit IgG	Thermo Fisher Scientific	A-21207	1:500

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
