# Peer review of "Fatty Acid Binding Protein 5 Mediates Cell Death by Psychosine Exposure through Mitochondrial Macropores Formation in Oligodendrocytes"

_biomedicines, 2020, doi:10.3390/biomedicines8120635_

Round 1
Reviewer 1 Report
Manuscript biomedicines-1012947 "Fatty acid binding protein 5 mediates cell death by psychosine exposure through mitochondrial macropores formation in oligodendrocytes". It is a comprehensive, very well conducted, and illustrated study on the mechanism of the cytotoxicity of psychosine in the oligodendrocytes. The small molecule FABP5 inhibitor's preclinical findings are of particular importance for further study of potential treatment of demyelinating diseases. It would be exciting to see in future research if the author is in vitro findings can translate to the in vivo models, i.e., twitcher mice or SOD1 mutant mice.
Minor criticism:
- The manuscript is missing an ethical permit number for the animal experiments protocol as well as the statement of compliance of use of animal in research, i.e., "Animal Research Advisory Committee Guidelines - NIH" or "institutional animal care and use committee - IACUC."
- The method of euthanasia is not described based on the pup's age and brain isolation, presumably decapitation, but it needs to be indicated.
- Whole western blot membrane images should be included in the supplementary data with the main manuscript's cutouts indicated.
- Immunoblotting analysis – which gel documentation system authors used for membrane detection? Please state.
- Figure 1D – authors indicate the mitochondria numbers per cell as non-significant between the group's Cont and AA; however, other groups that are presumably also not significant are not labeled. For style consistency, either label all groups or only present significant differences and state lack of significance for other groups in the figure legend. The same applies to all the figures.
- Figure 2,3 and 6 – the artworks are beautiful and informative. Is it an original artwork, or was it sourced from somewhere else? In the latter case, the source of the artwork needs to be stated.
- "Confocal microscopy of TOM20 (green) and DAPI (blue) (right), showing mitochondrial loss after psychosine treatment." – The purpose of the TOM20/DAPI staining panel is unclear. The same results are presented in panel H along with (dsDNA, FABP5) also, and the panel is missing control.
- In figure 2D, the scale bar interferes with the cutout in the second from the bottom panel, suggesting changing the scale bar's location or using a smaller scalebar.
- The authors demonstrate a very creative use of waifu2x-caffe software. The image resolution is always creating difficulties when studying mitochondria as the targets' size on the edge of the confocal microscope capabilities. Please provide the link to the waifu2x-caffe build used in the study. Also, since waifu2x is a neural network-based deconvolution algorithm, the statement of the limitation is required to address the fact that the algorithm may misinterpret and thus misrepresent small details during the denoise/upscale process. Waifu2x primary purpose is not upscaling of confocal images but drawn art, after all.
- Dot blot assay –Authors need to revise sentences in Dot blot assay description as currently, it states, "Membranes were then subjected to western blotting as mentioned above." I understand the intended meaning of the antibody incubation, but it may be confusing to the readership. Also, which antibody or reagent was used for dsDNA detection in the ECL process, please clarify.
- Authors use many antibodies for WB, immunofluorescence, and IP. It might be easier to represent the antibodies as the table rather than stating in the text.
- Figure legend is missing from the supplementary figure 1. Even though it is added in the Supplementary material part, it is better to add figure legend to the file itself.
- Proofreading of the final text is desirable.
Author Response
Reviewer 1: Manuscript biomedicines-1012947 "Fatty acid binding protein 5 mediates cell death by psychosine exposure through mitochondrial macropores formation in oligodendrocytes". It is a comprehensive, very well conducted, and illustrated study on the mechanism of the cytotoxicity of psychosine in the oligodendrocytes. The small molecule FABP5 inhibitor's preclinical findings are of particular importance for further study of potential treatment of demyelinating diseases. It would be exciting to see in future research if the author is in vitro findings can translate to the in vivo models, i.e., twitcher mice or SOD1 mutant mice.
Minor criticism:
- The manuscript is missing an ethical permit number for the animal experiments protocol as well as the statement of compliance of use of animal in research, i.e., "Animal Research Advisory Committee Guidelines - NIH" or "institutional animal care and use committee - IACUC."
Ans: We added that statement in the method.
- The method of euthanasia is not described based on the pup's age and brain isolation, presumably decapitation, but it needs to be indicated.
Ans: We added that statement in the method.
- Whole western blot membrane images should be included in the supplementary data with the main manuscript's cutouts indicated.
Ans: We provided our uncut images of western blot in the supplementary data.
- Immunoblotting analysis – which gel documentation system authors used for membrane detection? Please state.
Ans: We added that statement in the method.
- Figure 1D – authors indicate the mitochondria numbers per cell as non-significant between the group's Cont and AA; however, other groups that are presumably also not significant are not labeled. For style consistency, either label all groups or only present significant differences and state lack of significance for other groups in the figure legend. The same applies to all the figures.
Ans: We corrected it.
- Figure 2,3 and 6 – the artworks are beautiful and informative. Is it an original artwork, or was it sourced from somewhere else? In the latter case, the source of the artwork needs to be stated.
Ans: Thanks a lot, the schematic drafts were drawn by Adobe Illustrator CS6 software originally by myself.
- "Confocal microscopy of TOM20 (green) and DAPI (blue) (right), showing mitochondrial loss after psychosine treatment." – The purpose of the TOM20/DAPI staining panel is unclear. The same results are presented in panel H along with (dsDNA, FABP5) also, and the panel is missing control.
Ans: In fig.3 A (right), we provided the image of TOM20 (green) and DAPI (blue) of KG-1C cells treated with psychosine (5µM) for 24 hours, and image suggested obvious mitochondrial loss but integrated cell nucleus. It is the maximum time for psychosine (5µM) treatment in KG-1C. That is the reason why we did not try a longer time for psychosine exposure. In fig.3 H, dsDNA (red) is normally located in mitochondria surrounded by the mitochondrial outer membrane protein TOM20 (green) in control cells. In psychosine-treated cells, dsDNA (red) was released into the cytosol from the mitochondria, where FABP5 (blue) migrated to the mitochondria with TOM20 (green). This result is important for our further detection in dot blot assay.
- In figure 2D, the scale bar interferes with the cutout in the second from the bottom panel, suggesting changing the scale bar's location or using a smaller scalebar.
Ans: We corrected it.
- The authors demonstrate a very creative use of waifu2x-caffe software. The image resolution is always creating difficulties when studying mitochondria as the targets' size on the edge of the confocal microscope capabilities. Please provide the link to the waifu2x-caffe build used in the study. Also, since waifu2x is a neural network-based deconvolution algorithm, the statement of the limitation is required to address the fact that the algorithm may misinterpret and thus misrepresent small details during the denoise/upscale process. Waifu2x primary purpose is not upscaling of confocal images but drawn art, after all.
Ans: We provided the link to the waifu2x-caffe in figure legends.
- Dot blot assay –Authors need to revise sentences in Dot blot assay description as
currently, it states, "Membranes were then subjected to western blotting as mentioned above." I understand the intended meaning of the antibody incubation, but it may be confusing to the readership. Also, which antibody or reagent was used for dsDNA detection in the ECL process, please clarify.
Ans: We corrected that description.
- Authors use many antibodies for WB, immunofluorescence, and IP. It might be easier to represent the antibodies as the table rather than stating in the text.
Ans: We created a table for antibodies been used in this work.
- Figure legend is missing from the supplementary figure 1. Even though it is added in the Supplementary material part, it is better to add figure legend to the file itself.
Ans: We added the figure legend for supplementary figure 1.
- Proofreading of the final text is desirable.
Ans: We corrected it. The English in the manuscript was edited by a native speaker of Editage Co Ltd.
Reviewer 2 Report
The manuscript describes the role of fatty acid binding protein 5 (FABP5) in psychosine treated oligodendrocytes. The authors show that psychosine induces the expression of FABP5 in the oligos and through VDAC-1 and Bax mediate cell death. This is achieved by the generation mitochondrial macropores in the cell. The manuscript is well written and provide strong evidence for the hypothesis. Overall the studies are well done that data support the conclusions. Several issues need to be addressed.
1) Quantification of the data in figure 3 is needed.
2) More information is needed on the FABP5 inhibitors. Doses, where were they purchased or how synthesized.
Author Response
Reviewer 2: The manuscript describes the role of fatty acid binding protein 5 (FABP5) in psychosine treated oligodendrocytes. The authors show that psychosine induces the expression of FABP5 in the oligos and through VDAC-1 and Bax mediate cell death. This is achieved by the generation mitochondrial macropores in the cell. The manuscript is well written and provide strong evidence for the hypothesis. Over all the studies are well done that data support the conclusions. Several issues need to be addressed.
1) Quantification of the data in figure 3 is needed.
Ans: VDAC-1 oligomers were quantified in fig.3 B. ds-DNA levels in dot blot assay were quantified in fig.3 J.
2) More information is needed on the FABP5 inhibitors. Doses, where were they purchased or how synthesized.
Ans: Thank you very much for your kind suggestion, we added that statement in the method. The FABP5 inhibitors were synthesized by SHIRATORI PHAMACEUTICAL CO., LTD as described previously (13).
Reviewer 3 Report
Cheng et al. have found that fatty acid binding protein 5 (FABP5) expressed in psychosine-treated oligodendrocytes accelerates mitochondria-induced glial death by inducing mitochondrial macropore formation through voltage-dependent anion channels (VDAC-1) and BAX. The work is novel, and may be of interest for understanding Krabbe's disease. However, an important control is missing. The authors add 5 μM psychosine, but we do not know any of the details. Is psychosine dispersed in water? In buffer? Any detergent or other agent to help dispersion? Control experiments with vehicle only?
What is the amount of psychosine per cell, and/or per mg cell protein, and/or per mol of lipid phosphorus?
How do the authors know how much of the externally added psychosine is actually incorporated into the cell? Part of it will go into the aqueous medium, and a large part is likely to partition into the plasma membrane. Some kind of experimental approach is required to quantify the amount of psychosine actually entering the cytosol.
Minor points:
a) The structure and a few details of psychosine properties should be given in the Introduction. The authors could read Biochim Biophys Acta Biomembr. 2018 Dec;1860(12):2515-2526.
b) Krabbe and krabbe are used throughout the paper. The capitalised form should be uniformly used.
Author Response
Reviewer 3: Cheng et al. have found that fatty acid binding protein 5 (FABP5) expressed in psychosine-treated oligodendrocytes accelerates mitochondria-induced glial death by inducing mitochondrial macropore formation through voltage-dependent anion channels (VDAC-1) and BAX. The work is novel, and may be of interest for understanding Krabbe's disease. However, an important control is missing. The authors add 5 μM psychosine, but we do not know any of the details. Is psychosine dispersed in water? In buffer? Any detergent or other agent to help dispersion? Control experiments with vehicle only?
Ans: In this work, psychosine was first dissolved in DMSO and then mixed in the medium at a DMSO concentration 0.1% of the medium as described previously (J Lipid Res. 2006, 47(7):1478-92.). Under this condition, psychosine was dissolved in the medium.
What is the amount of psychosine per cell, and/or per mg cell protein, and/or per mol of lipid phosphorus?
Ans: It is an important issue to get to know how lipid molecule, psychosine works in the cell, protein and lipid. Psychosine is a lipid derived form galactosylceramide in the Krabbe disease and is easyly penetrated through cell membranes. We did not measure the concentration so far inside cell after psychosine treatment in this work, psychosine still triggered obvious mitochondria loss and cell death. It is enough for our further work to understand how FABP5 works in mitochondria under mitochondrial stress.
How do the authors know how much of the externally added psychosine is actually incorporated into the cell? Part of it will go into the aqueous medium, and a large part is likely to partition into the plasma membrane. Some kind of experimental approach is required to quantify the amount of psychosine actually entering the cytosol.
Ans: The concentration of psychosine we used in this work is according to the previous study (J Lipid Res. 2006, 47(7):1478-92.) in that psychosine (5μM) induced significant lysophosphatidylcholine (LPC) accumulation in primary oligodendrocyte. Thus, in our opinion, psychosine is incorporated into cells and function well.
Minor points:
- a) The structure and a few details of psychosine properties should be given in the Introduction. The authors could read Biochim Biophys Acta Biomembr. 2018 Dec;1860(12):2515-2526.
Ans: We corrected it in the introduction.
- b) Krabbe and krabbe are used throughout the paper. The capitalised form should be uniformly used.
Ans: Thank you very much for your kind suggestion, we corrected it.
Round 2
Reviewer 3 Report
I am astonished that I ask for major changes in a manuscript and I receive the "amended" version in less than 24 h. I take this as a disrespect from the authors. I was asking for a serious work to improve the paper,